# Research on the Duality of China’s Marine Fishery Carbon Emissions and Its Coordination with Economic Development

**DOI:** 10.3390/ijerph20021423

**Published:** 2023-01-12

**Authors:** Huanhuan Xiong, Xuejing Wang, Xinrui Hu

**Affiliations:** 1Research Center of the Central China Economic Development, Nanchang University, Nanchang 330031, China; 2School of Economics and Management, Nanchang University, Nanchang 330031, China

**Keywords:** marine fishery, carbon emissions, carbon sink, decoupling and coupling, coordination

## Abstract

Through the Tapio model, this paper measures the “decoupling and coupling” relationship between carbon emissions, carbon sinks, and economic growth of marine fisheries in nine coastal provinces of China in 2009–2019, objectively evaluates the economic benefits of carbon emissions and low-carbon development potential, and then discusses the economic development models of marine fisheries in detail. The results showed that the total carbon sink and carbon emission of China’s marine fisheries are increasing. Guangdong is dominated by “double low” economic benefits and low-carbon potential, and “double high” provinces have better resource endowment conditions; China’s marine fishery economic development is still dominated by conventional types. To further promote the sustainable development of China’s marine fisheries, all provinces should define the development orientation of marine fisheries, improve the production methods of marine fisheries according to local conditions, and adjust the industrial structure of marine fisheries in a timely manner, to achieve the low-carbon fishery goal of reducing carbon emissions and growing the economy.

## 1. Introduction

Global warming caused by carbon emissions has seriously affected the development of human society and economies, and emission reduction has become the focus of worldwide attention [1]. To deal with global climate change, we need to promote the reduction of greenhouse gas emissions, mainly carbon dioxide. All countries are reducing greenhouse gas in the form of global agreements [2]. In 2020, China proposed for the first time that carbon emissions should peak by 2030, and they will strive to be carbon neutral by 2060. The National “Fourteenth Five Year Plan” clearly proposed the concept of optimizing the green offshore aquaculture layout, building a marine ranching, and developing sustainable pelagic fisheries. In the background of vigorously promoting low-carbon marine economy, marine fisheries are an important part of the marine economy, and its sustained and healthy development is crucial. Marine fisheries have dual characteristics of “carbon source” and “carbon sink” in production activities. According to the definition of carbon source and carbon sink in the United Nations Framework Convention on Climate Change, the fuel consumption of fishing vessels engaged in marine fishing is an important source of carbon emissions in marine fisheries, namely “carbon source” [3], while “carbon sink” refers to the processes and mechanisms of promoting aquatic organisms to absorb carbon in water bodies through fishery production activities, and removing the carbon that has been converted into biological products from water bodies through fishing. Fishery production activities with carbon sink functions usually do not need bait, such as shellfish and algae cultivation in marine fisheries [4]. The introduction of the OECD decoupling indicator and Tapio decoupling elastic coefficient provides a new idea for studying the relationship between carbon emissions and carbon sinks and the economy, and also provides a reference for finding the answer to how to achieve the coordinated development of the economy and environment [5,6,7].

The research on carbon emissions from marine fisheries mainly focuses on three aspects. The first aspect is research on the sources of carbon emissions. Tang et al., believed that marine fishing is a high energy-consuming industry in the marine economy, and analyzed the reasons for high energy consumption in marine fishing [8]. Yue et al., concluded that the greenhouse gas emissions from marine fishing and fisheries showed a steady growth trend in combination with the energy consumption intensity, operation mode structure, and fuel emission coefficient of the operation mode [9]. The second aspect is achieving sustainable development of marine fisheries. Scholars studied how to reduce carbon emissions from marine fisheries. Friederike et al., pointed out that the training level of fishermen will have an important impact on carbon emissions of fisheries [10]. Zhang et al., pointed out the problems in China’s current fishery energy conservation and emission reduction work, and put forward suggestions to solve them [11]. Xu et al., pointed out that the main areas of reducing fishery energy consumption are trawling and gill net fishing boats, and the main technologies to promote fishery energy conservation and emission reduction are fishing boat technologies and recirculating aquaculture technologies [12]. The third aspect is the research on the driving factors of carbon emissions from marine fisheries. Shao et al., used the LMDI decomposition method to analyze the driving factors of carbon emissions from marine fisheries in China, and believed that energy intensity, carbon emission coefficient, industrial structure, industrial scale, etc., are all driving factors of carbon emissions [13].

At present, most scholars choose a specific perspective from which they conduct research on marine fishery carbon sinks. First, they define the marine fishery carbon sink. Tang et al., introduced the definition of marine fishery carbon sink and proposed countermeasures and suggestions for marine fishery carbon sink expansion [14]. The second is to study the relationship between carbon sinks and carbon emissions of marine fisheries. Zhu et al., used the LMDI decomposition method to calculate the net carbon emissions of marine fisheries in Zhejiang Province, and concluded that industrial scale, employee scale, and energy intensity are the three main factors affecting the net carbon emissions of marine fisheries in Zhejiang Province [15]. The third is to study the factors that affect the carbon sequestration capacity of marine fisheries. To measure the carbon sequestration capacity of marine fisheries in China’s coastal provinces, Ren et al., analyzed the factors that affect the carbon sequestration capacity of marine fisheries in various regions by using the logarithmic average Dickens indicator decomposition method [16]. Yang et al., used a method based on the LMDI decomposition method, to deconstruct the contributing factors to the improvement of China’s marine fishery carbon sequestration capacity, and predict the carbon sequestration potential of China’s marine fisheries in 2030 under different scenarios [17].

With the development of the marine economy restricted by natural resources, the interaction between marine fisheries and economic development has gradually become a new hotspot in the marine field. At present, most scholars focus on the relationship between marine fisheries and economic development. Wang et al., used the panel threshold model to analyze the impact of changes in the marine fishery industrial structure on marine fishery economic growth in a multi-dimensional way [18]. Sun et al., analyzed the economic efficiency of marine fisheries in coastal provinces and urban areas in combination with the three-stage DEA model [19]. Han et al., used an entropy topsis model and super efficiency SBM model to measure the resilience and efficiency of China’s marine fishery economy, respectively, and described the co-evolution characteristics of the two through the Haken model [20].

The above research results provide a theoretical basis and reference that is significance for this paper [21,22,23,24,25]. However, due to the wide range of marine fisheries, it is difficult to unify accounting methods and the accuracy of data is not high. With China’s increasing support for the economic development of marine fisheries and the deepening of the integration of marine fisheries with other industries, carbon emissions from marine fisheries have become an important target for emission reduction. Moreover, there is a typical difference between the marine fisheries and other industries. There are a large number of emission sources and a large number of absorption sources. It is a dual industry. Therefore, how to reduce the emission sources and increase the carbon sink of marine fisheries in the process of emission reduction has become a problem worth studying. At the same time, due to China’s vast territory, the progress of the fishery economy in different regions is quite different. Studying the duality of the fishery economy and the coordination of fishery economic development in different regions will provide a basis for summarizing the development mode of fishery economy and promoting the development direction of low-carbon fisheries with regional differentiation.

## 2. Data Sources and Research Methods

### 2.1. Data Sources

Marine fisheries are one of the representative industries with high energy consumption in the marine economy. Therefore, in this paper, when estimating the carbon emissions of marine fisheries in the nine coastal provinces of China (see Figure 1), the calculation is mainly based on the operating mode power of marine fishing vessels. The estimation of marine carbon sink is mainly calculated using the output of marine shellfish aquaculture. China’s marine aquaculture can be divided into fish, shellfish, crustaceans (shrimp, crab), algae, seafood, etc. More than 95% of the output in the research of fishery carbon sinks has been invested in the production process, and fish and crustaceans do not belong to the category of fishery carbon sinks [26,27,28]. Therefore, this study defines marine aquaculture carbon sinks as shellfish and algae carbon sinks. China’s marine cultured shellfish and algae consists mainly of clams, oysters, scallops, mussels, kelp, seaweed, wakame, and gracilaria, and the rest were classified into other categories for calculation. The data on the operation mode of fishing boats, the data on the production of shellfish and algae in marine aquaculture, and the data on the economic output value of oceanic fisheries are all from the “China Fishery Statistical Yearbook” (2009–2019).

### 2.2. Carbon Emission Measurement Methods

The calculation process of carbon emissions refers to the calculation method of carbon dioxide emissions from fossil fuel combustion proposed by the ORNL [29]. The calculation formula is as follows:(1)$${\;C}_{\mathrm{CO}2}=P_{\mathrm{fuel}}\times h\times k\times n\times \xi \times \beta$$

In Formula (1): $C_{\mathrm{CO}2}$ is the carbon released from marine fishing boats; P_fuel_ is the fuel oil consumption of marine fishing boats; h is the fuel oil conversion coefficient of standard coal 1.4571; k is the effective oxidation fraction of 0.982; n is the content of standard coal per ton (the carbon content is 0.73257); ξ is the ratio of carbon dioxide emissions from fuel oil to carbon dioxide emissions from coal combustion under the condition of obtaining the same thermal energy, which is a constant of 0.813; and β is based on the relative atomic mass, and the carbon-converted carbon dioxide constant is 44/12, which is approximately equal to 3.67.

In addition, when calculating the fuel consumption P of marine fishing vessels, according to the different types of operations in the marine fishing industry, this paper estimated the annual marine fishing vessels according to the “Reference Standard for the Measurement of Fuel Subsidy for Domestic Motorized Fishing Vessels” issued by the Ministry of Agriculture and Rural Affairs. The specific coefficient of fuel consumption is shown in Table 1:

### 2.3. Calculation Method for Carbon Sink

According to the carbon sequestration mechanism of shellfish and algae, the carbon fixed by shellfish is contained in shells and soft tissues, so the carbon sequestration of shellfish in the mariculture industry is the sum of carbon content in shells and soft tissues; the carbon fixed by algae is the carbon content of algae. The calculation methods and accounting coefficients of the carbon sinks of shellfish and algae were obtained by referring to Tang et al. [8], Zhang et al. [11], Shao et al. [13], Yue et al. [30], and Ji et al. [31] (see Equations (2)–(6), Table 2).
(2)Marine aquaculture carbon sink=shellfish carbon sink+ algae carbon sink
(3)Shellfish carbon sink= soft tissue carbon sink + shell carbon sink
(4) Soft tissue carbon sink=shellfish yield × dry−wet coefficient ×Soft tissue proportion× soft tissue carbon content
(5)Shell carbon sink=shellfish yield × dry−wet coefficient × shell proportion × shell carbon content
(6)Algae carbon sink=algae yield×dry−wet coefficient× carbon content

### 2.4. Decoupling Coupling Model

Based on the OECD decoupling indicator and the Tapio decoupling elastic coefficient, the decoupling evaluation model and coupling evaluation model in this paper are as follows [32,33]:(7)$$\;T_{it}\left(E,A\right)=\frac{\left(E_{i}-E_{i-1}\right)/E_{i-1}}{\left(A_{i}-A_{i-1}\right)/A_{i-1}}$$
(8)$$T_{io}\left(C,G\right)=\frac{\left(C_{i}-C_{i-1}\right)/C_{i-1}}{\left(G_{i}-G_{i-1}\right)/G_{i-1}}$$

Among them: *T_it_* is the economic growth elasticity of marine fishery carbon emissions; *E_i_* is the carbon emissions of marine fisheries; *A_i_* is the economic output value of marine fisheries; *T_io_* is the economic growth elasticity of the marine fishery carbon sink; *C_i_* is the marine fishery carbon sink; and *G_i_* is the economic output value of the marine fishery. According to the value of the indicator and its economic meaning, the eight decoupling states and coupling states are divided into three categories: environmental protection type, conventional type, and pollution type, as shown in Table 3 and Table 4.

## 3. Analysis of Marine Fishery Carbon Emissions/Sinks

### 3.1. Analysis of Carbon Emissions from Marine Fisheries

It can be seen from Table 5 and Figure 2 that the total carbon emissions of China’s marine fisheries increased year by year to a peak of 19.4408 million tons before 2015, and then decreased year by year. The average carbon emission is about 2.0332 million tons, showing a slight increase, but the difference in carbon emission between regions is relatively obvious. Zhejiang’s carbon emission ranks first, and Hebei’s carbon emission is the lowest. Average carbon emissions peaked during the period in 2015 and then declined slightly. The standard deviation of China’s marine fishery carbon emissions fluctuates between 1.3098 and 1.4099 million tons.

### 3.2. Analysis of Carbon Sinks in Marine Fisheries

As can be seen from Table 6 and Figure 3, the average value of carbon sinks in China’s marine fisheries from 2009 to 2019 was around 129,600 tons, showing slight fluctuations, and regional differences that were still expanding. Compared with other regions, the marine fishery carbon sequestration in Hainan is low, so the values shown in the figure are not clear. From 2009 to 2019, the carbon sink was around 1.1666 million tons. From 2009 to 2019, the carbon sink increased year by year, reaching a peak value of 1.3399 million tons in 2019. There were noticeable changes in the maximum and minimum carbon sinks, showing an increase first and then a decrease changing trend. The standard deviation of marine fishery carbon sinks generally showed an increasing tendency, expanding from 9.23 in 2009 to 13.12 in 2018.

### 3.3. Analysis of Net Carbon Emissions from Fisheries

From Table 7, during the period of 2009–2019, the net carbon emissions of the nine coastal provinces showed a trend of first increasing and then decreasing. As a whole, there was little fluctuation, and the growth and decline rates were not obvious. Net carbon emissions of the marine fisheries in Zhejiang were higher than those in other regions, mainly because the carbon emissions of the marine fisheries in Zhejiang were significantly higher than those in other regions, and Zhejiang’s carbon sinks were lower than those in other regions. The net carbon emission of the marine fisheries in Zhejiang was over 3 million tons, and net carbon emission of the marine fisheries in other regions except Zhejiang was below 3 million tons, which may be related to the formulation of economic policies in the different regions.

### 3.4. Study on the Influencing Factors of Marine Fishery Economic Development

#### 3.4.1. Multiple Regression Analysis Model and Indicator Construction

The regression analysis forecasting method is based on the analysis of the relationship between independent variables and dependent variables of market phenomena, and establishes the regression equation between variables. This method uses the regression equation as a forecasting model to predict the relationship between dependent variables according to the quantitative changes of independent variables in the forecasting period. It is divided into univariate and multivariate regression analysis forecasting methods, and there are many factors affecting economic development. This paper selected the multivariate regression analysis forecasting method. According to the different correlation between independent variables and dependent variables, it can be separated into linear regression prediction and nonlinear regression prediction. This paper analyzed several important factors affecting the development of marine fishery economy in the nine coastal provinces from the perspective of multiple linear regression prediction.

The multiple linear regression prediction model is a linear regression model with multiple explanatory variables, and its equation can be written as:(9)$$Y=\beta_{0}+\beta_{1}X_{1}+\beta_{2}X_{2}+\ldots +\beta_{P}X_{P}+\in$$
where β_0_ is the regression constant, and β_1_, β_2_, …, β_p_ are the overall regression parameters. When p = 1, the formula is a linear regression model with one variable; when p ≥ 2, the formula is a multiple linear regression model and ε is random error and ε~N (0, σ^2^).

The data used in this analysis comes from the China Fishery Statistical Yearbook from 2009 to 2019. There are many factors that affect the development of marine fishery economy, and the relationship between each factor and the development of the marine fishery economy is complicated. Marine fishery output will affect the development of the marine fishery economy, and marine fishery output is also one of the factors to verify the healthy development of marine fishery economy [34,35]. Environmental regulation will affect the formulation of economic policies, and different economic policies have an important impact on economic development [36,37,38]. The amount of marine fishery carbon sinks will affect the economic value of marine fishery carbon sinks, thus indirectly affecting the economic development of marine fisheries [39,40,41,42,43]. Marine technological innovation reduces the production cost by reducing the input of production factors such as labor force. Technology, as a new element, can be continuously added or innovated to achieve the purpose of reducing costs and increasing efficiency. Therefore, people have paid more and more attention to the influence of technological innovation on the development of the marine fishery economy [44,45]. Fisheries have a double impact on the carbon dioxide content in the atmosphere, and excessive pursuit of production capacity will lead to environmental crisis, thus causing huge economic losses [46]. In this paper, the factors influencing the development of marine fishery economy are comprehensively discussed from five aspects: marine fishery output, environmental regulation, marine fishery carbon sink, technological innovation, and marine fishery carbon emission. According to the principles of generality, particularity, importance, and availability of indicators, this analysis selected five representative indicators with available data. The total output value of a certain area or industry is often of great significance to economic development, and it is measured by the economic output value [46]. The marine fishery output comprehensively reflects the fishery resources and fishery output capacity of a region, and it is of great significance to the development of the fishery economy, so the marine fishery output indicator was selected [47,48,49]. Environmental regulation is measured by pollution economic loss [50,51]. The indicator for marine fishery carbon sink is the amount of marine fishery carbon sink [52,53]. Marine fishery science and technology innovation reflects the technical level of science and technology. In recent years, the proportion of science and technology in the development of the marine fishery economy is getting higher and higher; thus, marine fishery science and technology innovation is an important part of the development level of the marine fishery economy [54]. Technology innovation is measured by technical investment [55,56]. Marine fishery carbon emissions will affect economic development, so the marine fishery carbon emissions indicator was selected [57,58,59]. Following the existing literature. The economic output value was determined as dependent variable Y, and marine fishery output, pollution economic loss, carbon sink, technical investment, and carbon emission were determined as X_1_, X_2_, X_3_, X_4_, and X_5_, respectively. Thus, the multivariate linear regression prediction model of y can be obtained:(10)$$Y=\beta_{0}+\beta_{1}X_{1}+\beta_{2}X_{2}+\beta_{3}X_{3}+\beta_{4}X_{4}+\beta_{5}X_{5}+\in$$

#### 3.4.2. Descriptive Statistical Analysis

As can be seen from the Table 8, the average output of marine fisheries was 17,780,200 tons, the minimum was 14,038,200 tons, and the maximum was 20,648,100 tons. The average carbon sink of marine fisheries was 1,166,000 tons, the minimum was 929,100 tons, and the maximum was 1,339,900 tons. The average carbon emission of marine fisheries was 18,298,700 tons and the minimum was 16,923,800 tons.

#### 3.4.3. Correlation Analysis

Correlation analysis is a statistical method to measure the close correlation between multiple explanatory variables and explained variables. Through the correlation analysis in Stata software, we can find the correlation between independent variables and the correlation between independent variables and dependent variables. Table 9 shows the correlation coefficient between variables, including Pearson correlation coefficients between six variables including independent variables and dependent variables. Among them, the correlation coefficients of marine fishery output, marine fishery carbon sink, technical investment, and economic output value of marine fisheries were all greater than 0.80, and the significant correlation coefficients were all less than 0.01, which indicates that they have a strong linear correlation. The correlation coefficient between pollution economic loss and the economic output value of marine fisheries was −0.44, indicating that there is no correlation between pollution economic loss and the economic output value of marine fishery.

#### 3.4.4. OLS Linear Regression Analysis

From the regression results in the Table 10 and Table 11, it can be seen that the F value measures the overall significance of the model. The value of F was 0.0000, which is less than 0.01, that is, the model was effective at the significance level of 0.01. The Adj R-squared value measures the fitting degree of the model, and the Adj R-squared value was 0.9942, which indicates that the fitting degree of the model is high. T represents whether each variable is significantly related to dependent variables, such as marine fishery output, marine fishery carbon sink, technical investment, and economic output value of marine fisheries.

## 4. Coordination Analysis of Marine Fishery Carbon Emissions and Marine Fishery Economic Development

### 4.1. Coordination Static Analysis

In Formulas (11) and (12), the economic benefit of carbon emission is the economic output value per unit of carbon emission; the low-carbon development potential is the ratio of carbon sink to carbon emission; U and F are the economic benefit of carbon emission and low-carbon development potential of each province and city, respectively; and u and f are the national carbon emission economic benefits and low-carbon development potential. According to the comparison of carbon emission economic benefits and low-carbon development potential with the national average level, the regions are divided into grades, as shown in Table 12.
(11)carbon emission economices=carbon emission economic output/carbon emission
(12)low carbon development potential=carbon sink/carbon emission

From Table 13, it can be seen that in the past 10 years, 44.44% of China’s provincial areas have changed the level of fishery carbon emission economic benefits and low-carbon agriculture development potential. Among them, 25.00% of the provincial areas have improved economic benefits, and the economic benefit level has mainly decreased. In Fujian Province, only 22.22% of the province’s low-carbon potential grades have changed, while Shandong Province maintains a high level of economic benefits and low-carbon potential. In 2009, the regions with high economic efficiency mainly included Fujian, Shandong, and Hainan, while Liaoning, Jiangsu, Guangxi, Shandong, and Fujian showed a high level of low-carbon development potential as a whole. In 2019, only the economic benefit level of Zhejiang increased, and the economic benefit level of Fujian in the eastern coastal areas decreased. The low-carbon potential of Hebei region has increased, and the low-carbon potential of Jiangsu region has decreased. Shandong Province is a major marine fishery province in China and one of the important fishery production bases in China. The output of kelp cultures accounts for more than 50% of the total aquaculture output in China. In February, 2011, the State Council officially approved the Plan of Zhejiang Marine Economic Development Demonstration Zone, which marked that the construction of the Zhejiang Marine Economic Development Demonstration Zone was promoted to a national strategy. As a result, the development of the marine fishery economy in Zhejiang Province entered the fast lane, made great progress, and played a leading role in the whole country. Fujian Province benefited from the national policy earlier. In 2010, it implemented preferential measures of fishery policy insurance, such as increasing the insurance amount for fishing boats, increasing financial subsidies, and expanding the scope of pilot projects, which promoted the expansion of the mariculture area in Fujian Province, enhancing its carbon sequestration capacity, and its economic value was at the forefront. The planning and implementation of the Guangdong Marine Economy Comprehensive Experimental Zone in 2012 laid a policy foundation for the development of the marine fishery economy.

### 4.2. Coordination Dynamic Analysis

#### 4.2.1. Phase Characteristics of the State of “Decoupling and Coupling” in Nine Coastal Provinces

It can be seen from Table 14 that the decoupling relationship between China’s marine fishery carbon emissions and economic development was dominated by the conventional type, and shows a certain periodic law. An ideal environmental protection model with economic growth showed a reduction in carbon emissions in 2018. From 2010 to 2015, it was a conventional development, and the growth rate of carbon emissions was slower than the economic growth rate, showing a weak decoupling state where the carbon emissions per unit of marine fisheries were less than 0.8. During the “Twelfth Five-Year Plan” period, the overall decoupling of marine aquaculture carbon sinks and economic development, and the state was relatively unstable. The economic benefits were higher than the environmental benefits, indicating that the marine fisheries are still in a state of high pollution and high energy consumption, and the overall quality of the fishery labor force is relatively low. From 2016 to 2019, the carbon emissions of fisheries experienced a process of reduction, while the decoupling type experienced a process of “environmental protection-pollution”. The coupling relationship between China’s fishery carbon sink changes and economic development is conventional and environmentally friendly. In the past 10 years, the average annual growth rate of China’s marine fishery economic output value reached 10%, but the carbon sink has only maintained an average annual growth rate of 3.8%. Except for 2015 and 2019, the carbon sink and economic development have maintained a weak coupling state. In 2015, the increase in carbon sinks in China was comparable to the economic growth rate, and the carbon sinks per unit of economic output value reached 0.95, showing a growth-coupling relationship. In 2019, China’s carbon sink increased faster than the economic growth rate, and the carbon sink per unit of economic output value reached 1.89, showing a strong coupling relationship.

#### 4.2.2. Characteristics of the “Decoupling” State of the Nine Coastal Provinces

It can be seen from Table 15 and Table 16 that in 2010, the economic development of marine fishery carbon emission in China’s coastal areas included environmental protection, conventional, and pollution types, while the development of the marine fishery carbon sink economy was dominated by conventional types. Among them, 33.33% of the provinces had a weak decoupling relationship between carbon emissions and economic development, and 22.22% of them had a strong decoupling relationship between carbon emissions and economic development. Only Jiangsu and Guangdong provinces had a type of pollution development with strong and negative decoupling. Guangxi and Hainan had a strong decoupling relationship. These provinces present the pollution development type of expansion and negative decoupling. Three types of fishery carbon sink economic development models in the nine provinces are equally important; 66.67% of the provinces are weakly coupled, and Jiangsu presents the pollution type. In 2019, the development of the marine fishery carbon emission economy in China’s coastal areas included environmental protection, conventional, and pollution types, while the development of marine fishery carbon sink economy was still dominated by conventional ones. The economic development of marine fishery carbon emission in China’s coastal areas has changed. Guangdong, Guangxi, and Hainan have changed from pollution to environmental protection, Hebei and Zhejiang have changed from environmental protection to conventional, and the proportion of environmentally friendly provinces has increased by 11.11%. The conventional economic development model of carbon sinks in China’s coastal areas has declined, and Hebei, Shandong, and Hainan have changed from conventional to pollution, and the proportion of conventional provinces has dropped by 33.33%. Only Liaoning Province has changed from an environmental protection type to a conventional type, and the carbon sink economic development model of Jiangsu, Zhejiang, Guangdong, and Guangxi has not changed.

#### 4.2.3. Evolution of the Coupled Development Model of Fishery Economy and Carbon Emission Decoupling in Nine Coastal Provinces

To better analyze the differences in the economic development models of the nine coastal provinces, combined with the type of decoupling coupling, the economic development of marine fishery carbon emissions is divided into ideal type, general type, and inferior type, as shown in Table 17.

From Table 18, it can be seen that from 2010 to 2019, the pattern of China’s fishery carbon emission economic development model changed to a certain extent. The proportion of provinces with lower grades accounted for 44.44%, namely Hebei, Zhejiang, Fujian, and Shandong, and the proportion of provinces with higher grades accounted for 22.22%. The change in carbon sink pattern is the dominant reason for the grade change. Specifically, in 2009, China’s fishery economic development model was 33.33%, mainly in Hebei, Fujian, and Shandong. The inferior development provinces were mainly Jiangsu, Guangdong, Guangxi, and Hainan. In 2019, the ideal economic development model of China’s fishery accounted for 22.22%, the same as in 2010, and the inferior economic development model accounted for 55.55%, an increase compared to 2010.

## 5. Conclusions

In this paper, the carbon emissions and sinks of marine fisheries in nine coastal provinces from 2009 to 2019 and the factors affecting the economic development of marine fisheries were analyzed. Combined with the changes in economic development in the nine coastal provinces, the decoupling relationship between carbon emissions and sinks of oceanic fisheries and economic development was obtained. Finally, the following conclusions were drawn:

(1) Depending on the changes of carbon emissions from marine fisheries, there are obvious differences in carbon emissions among the nine coastal provinces. Carbon emissions of marine fisheries in Zhejiang rank first, and the carbon emissions of marine fisheries in other areas except Zhejiang are less than 3 million tons. It can be seen from the changes of the carbon sink in marine fisheries that province with the highest carbon sink in marine fisheries was Shandong, with an average of 300,000 tons, followed by Fujian > Guangdong > Liaoning > Zhejiang > Guangxi > Jiangsu > Hebei > Hainan. The net carbon emission of fisheries showed that the net carbon emission of Zhejiang was greater than that of other regions, and that of Hebei was the lowest. Zhejiang has long since implemented fishery development, and Zhoushan Fishing Ground is located here. The operating base of fishing boats is ahead of other provinces, and the scale of marine fishing is huge. There is plenty of room for improvement in reducing emissions and increasing foreign exchange, or improving fishery technology.

(2) According to the data of economic benefits of marine fishery carbon emissions, the economic benefits of Jiangsu, Fujian, Shandong, and Hainan were at a high level, indicating that the economic benefits of unit carbon emissions are high. With the economic development, the carbon emissions of marine fishery were also high. From the change of low-carbon development potential, Liaoning, Jiangsu, Fujian, Shandong, and Guangxi had a high level of low-carbon development potential. From the analysis of the factors that affect the development of the marine fishery economy, marine fishery output, marine fishery carbon sink, and technical investment will all affect the progress of the marine fishery economy.

(3) According to the calculation of carbon emissions and carbon sinks of marine fisheries, combined with the economic output value data of the nine coastal provinces, the decoupling model of carbon emissions, carbon sinks, and economic development was constructed, the decoupling indicator was calculated, and the decoupling relationship was analyzed. The decoupling values of Hebei, Zhejiang, and Guangdong were low, indicating that these areas are developing low-carbon economies while taking into account the environment. In general, the relationship between carbon emissions from marine fisheries and economic development in the nine coastal provinces was obvious, and the decoupling state was relatively good.

## 6. Recommendations

The nine coastal provinces are in the rising stage of social and economic development, which will inevitably produce carbon dioxide emissions. This study analyzed the decoupling relationship between marine fishery carbon emissions, carbon sinks, and economic development, and based on the research results, combined with the actual situation of the fishery economy in the various regions, we put forward the following measures and suggestions to reduce emissions, increase sinks, and develop green low-carbon economy:

(1) Increase shellfish culture, systematically optimize the variety structure of shellfish, and promote the promotion of ecological and economic benefits for marine fisheries. At present, due to the distinct breed and unreasonable structure, the ecological and economic benefits of marine fisheries in China need to be improved. For a long time, the absolute quantity and growth rate of shellfish culture are the largest. To improve the comprehensive benefits of the marine fishery eco-economy in China, we should pay heed to the adjustment, optimization, and promotion of marine aquaculture variety structure to realize the sustainable economic growth of marine fisheries.

(2) Improve the carbon emission policy and promote the development of carbon emission trading. Using policy and legal means to formulate binding indicators and management regulations of carbon emissions from marine fisheries to ensure the smooth realization of low-carbon utilization management objectives of marine fisheries is conducive to the smooth progress of low-carbon work in marine fisheries. These nine provinces need to improve the carbon trading market system, strengthen the construction of a carbon trading system, and define the main scope of carbon emissions.

(3) The economic development of marine fishery should consider the technological innovation of regional marine fisheries. Technological innovation will affect the economic development of marine fisheries in various regions. Increase investment in marine fishery technology, accelerate the advancement of the marine fishery economy, and make China’s marine fisheries a new economic growth point. Choose the province with strong technological innovation from the nine coastal provinces to build a new marine fishery economic system.

## Figures and Tables

**Figure 1 ijerph-20-01423-f001:**
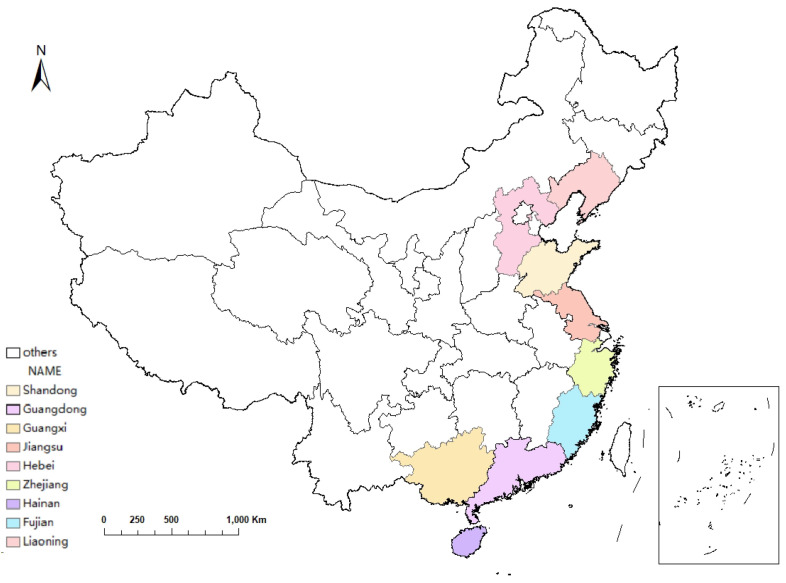
Location of nine provinces of China in 2022.

**Figure 2 ijerph-20-01423-f002:**
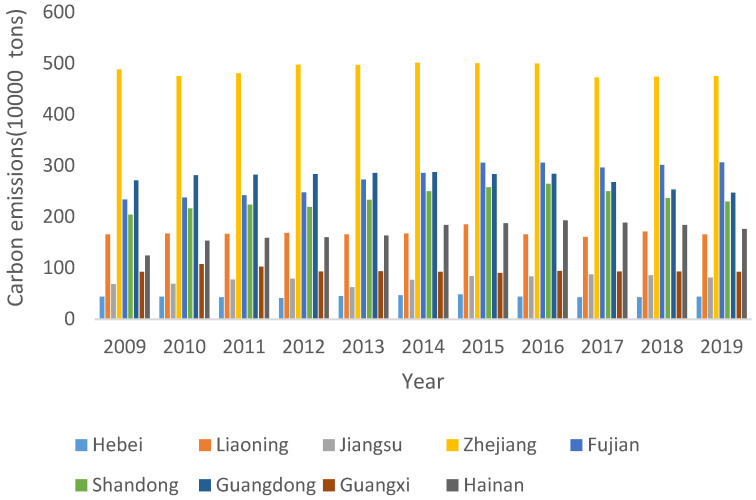
Carbon emissions from China’s marine fisheries from 2009 to 2019.

**Figure 3 ijerph-20-01423-f003:**
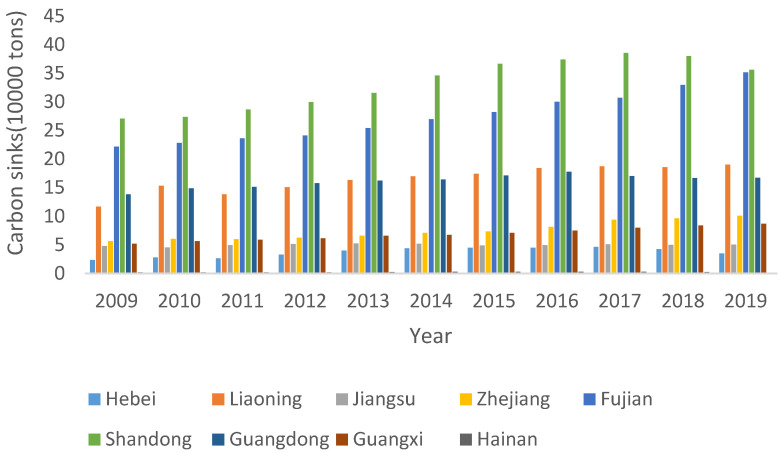
Carbon sinks from China’s marine fisheries from 2009 to 2019.

**Table 1 ijerph-20-01423-t001:** Subsidy oil consumption coefficient of marine fishing vessels by operation type (t/Kw).

Indicator	Trawl	Purse Seine	Gill Net	Trap Net	Fishing Tackle	Others
Coefficient	0.480	0.492	0.451	0.328	0.328	0.321

**Table 2 ijerph-20-01423-t002:** Accounting coefficients of carbon sink capacity of mariculture (%).

Species	Wet and Dry Coefficient	Proportion of Soft Tissue Mass	Percentage of Shell Mass	Soft Tissue Carbon Content	Carbon Content of Shells
Clam	52.55	1.98	98.02	44.90	11.52
Scallops	63.89	14.35	85.65	42.84	11.40
Oysters	65.10	6.14	93.86	45.98	12.68
Mussels	75.28	8.47	91.53	44.40	11.76
Other shellfish	64.21	11.41	88.59	43.87	11.44
Kelp	20.00	1.00	0.00	31.20	0.00
Wakame	20.00	1.00	0.00	26.40	0.00
Nori	20.00	1.00	0.00	27.39	0.00
Gracilaria	20.00	1.00	0.00	20.60	0.00
Other algae	20.00	1.00	0.00	27.76	0.00

**Table 3 ijerph-20-01423-t003:** Decoupling indicator grade and elasticity value.

T_it_	∆A/A	Decoupling State	Development Type
(−∞, 0)	>0	Strong decoupling	Environmentally friendly
(0, 0.8)	>0	Weak decoupling	Conventional
[0.8, 1.2)	>0	Growth link	Conventional
[1.2, +∞)	>0	Expansion negative decoupling	Pollution type
(−∞, 0)	<0	Strong negative decoupling	Pollution type
(0, 0.8)	<0	Weak negative decoupling	Pollution type
[0.8, 1.2)	<0	Decline link	Conventional
[1.2, +∞)	<0	Recession decoupling	Conventional

**Table 4 ijerph-20-01423-t004:** Coupling indicator grade and elastic value.

T_io_	∆C/C	∆G/G	Coupling State	Development Type
(−∞, 0)	<0	>0	Strong decoupling	Pollution type
(0, 0.8)	>0	>0	Weak coupling	Conventional
[0.8, 1.2)	>0	>0	Growth coupling	Conventional
[1.2, +∞)	>0	>0	Strong coupling	Environmentally friendly
(−∞, 0)	>0	<0	Strong negative decoupling	Pollution type
(0, 0.8)	<0	<0	Weak negative coupling	Conventional
[0.8, 1.2)	<0	<0	Decay coupling	Conventional
[1.2, +∞)	<0	<0	Strong negative coupling	Pollution type

**Table 5 ijerph-20-01423-t005:** Carbon emissions from marine fisheries in nine coastal provinces from 2009 to 2019 year (10,000 tons).

Area	2009	2010	2011	2012	2013	2014	2015	2016	2017	2018	2019
Hebei	44.07	43.78	42.78	41.03	45.06	46.90	48.37	44.14	42.56	42.51	43.68
Liaoning	165.82	167.22	167.06	168.52	165.50	167.54	185.03	165.46	160.58	171.10	165.70
Jiangsu	68.32	69.03	77.66	78.99	62.29	76.92	84.16	83.29	87.64	85.92	81.59
Zhejiang	488.33	475.44	481.17	498.00	497.34	501.66	500.71	499.87	472.45	474.45	475.41
Fujian	234.10	237.85	242.19	247.78	272.97	285.79	305.99	306.28	296.65	301.75	306.84
Shandong	204.12	216.58	223.60	219.28	233.30	250.16	258.18	264.49	249.97	236.91	229.98
Guangdong	271.16	281.40	282.84	283.83	285.71	287.82	283.78	284.29	268.00	253.64	247.17
Guangxi	92.37	107.80	102.50	92.96	93.86	92.74	90.23	94.00	93.16	92.99	92.75
Hainan	124.09	153.52	158.93	159.92	163.23	184.01	187.63	193.05	188.51	184.32	176.49
Total	1692.38	1752.62	1778.74	1790.31	1819.26	1893.55	1944.08	1934.87	1859.51	1843.60	1819.60

**Table 6 ijerph-20-01423-t006:** Carbon sinks of marine fisheries in nine coastal provinces from 2009 to 2019 year (10,000 tons).

Year	2009	2010	2011	2012	2013	2014	2015	2016	2017	2018	2019
Hebei	2.37	2.79	2.63	3.30	4.02	4.41	4.50	4.50	4.65	4.23	3.48
Liaoning	11.68	15.33	13.83	15.07	16.35	16.98	17.46	18.45	18.73	18.57	19.05
Jiangsu	4.77	4.54	4.95	5.13	5.23	5.18	4.87	4.95	5.08	4.99	5.03
Zhejiang	5.66	6.02	6.01	6.22	6.57	7.08	7.36	8.14	9.37	9.62	10.11
Fujian	22.19	22.84	23.64	24.12	25.45	27.00	28.22	30.02	30.75	33.00	35.16
Shandong	27.07	27.40	28.66	30.00	31.58	34.62	36.69	37.43	38.59	38.03	35.61
Guangdong	13.82	14.87	15.16	15.80	16.23	16.43	17.16	17.81	17.02	16.69	16.74
Guangxi	5.17	5.66	5.87	6.12	6.58	6.73	7.11	7.49	7.97	8.41	8.67
Hainan	0.18	0.18	0.18	0.20	0.23	0.28	0.28	0.29	0.31	0.23	0.14
Total	92.91	99.63	100.92	105.96	112.24	118.72	123.64	129.07	132.48	133.76	133.99

**Table 7 ijerph-20-01423-t007:** Net carbon emissions of marine fisheries in nine coastal provinces from 2009 to 2019 (10,000 tons).

Year	2009	2010	2011	2012	2013	2014	2015	2017	2018	2019
Hebei	41.7	40.99	40.15	37.73	41.04	42.49	43.87	37.91	38.28	40.2
Liaoning	154.14	151.89	153.23	153.45	149.15	150.56	167.57	141.85	152.53	146.65
Jiangsu	63.55	64.49	72.71	73.86	57.06	71.74	79.29	82.56	80.93	76.56
Zhejiang	482.67	469.42	475.16	491.78	490.77	494.58	493.35	463.08	464.83	465.3
Fujian	211.91	215.01	218.55	223.66	247.52	258.79	277.77	265.9	268.75	271.68
Shandong	177.05	189.18	194.94	189.28	201.72	215.54	221.49	211.38	198.88	194.37
Guangdong	257.34	266.53	267.68	268.03	269.48	271.39	266.62	250.98	236.95	230.43
Guangxi	87.2	102.14	96.63	86.84	87.28	86.01	83.12	85.19	84.58	84.08
Hainan	123.91	153.34	158.75	159.72	163	183.73	187.35	188.2	184.09	176.35

**Table 8 ijerph-20-01423-t008:** Descriptive statistics.

Variable	Obs	Mean	Std. Dev.	Min	Max
Economic output value	11	7223.88	1576.14	8967.43	8990.20
Marine fishery output	11	1778.02	221.51	1403.82	2064.81
Pollution economic loss	11	114,137.5	136,668.20	25,933.5	509,507.2
Carbon sink	11	116.67	15.12	92.91	133.99
Technical investment	11	94,830.30	43,158.62	44,083.36	168,581.40
Carbon emission	11	1829.87	76.46	1692.38	1944.08

**Table 9 ijerph-20-01423-t009:** Correlation analysis.

	Economic Output Value	Marine Fishery Output	Pollution Economic Loss	Carbon Sink	Technical Investment	Carbon Emission
Economic output value	1					
Marine fishery output	0.988 ***	1				
Pollution economic loss	−0.440	−0.491	1			
Carbon sink	0.976 ***	0.996 ***	−0.518	1		
Technical investment	0.898 ***	0.943 ***	−0.468	0.938 ***	1	
Carbon emission	0.779 ***	0.740 ***	−0.346	0.745 ***	0.485	1

Robust t-statistics in parentheses *** *p* < 0.01.

**Table 10 ijerph-20-01423-t010:** Multiple linear regression analysis.

Source	SS	df	MS	Number of Obs	=	11
				F(5, 5)	=	344.36
Model	2.7247 × 10^15^	5	5.4495 × 10^14^	Prob > F	=	0.0000
Residual	7.9125 × 10^12^	5	1.5825 × 10^12^	R-squared	=	0.9971
				Adj R-squared	=	0.9942
Total	2.7326 × 10^15^	10	2.7326 × 10^14^	Root MSE	=	1.3 × 10^6^

**Table 11 ijerph-20-01423-t011:** Multiple linear regression analysis.

Economic Output Value	Coef.	Std. Err.	t	P > |t|	[95% Conf.]	[Interval]
Marine fishery output	1825.35	24,487.23	7.45	0.001	119,588.6	245,481.4
Pollution economic loss	3.86431	3.681944	1.05	0.342	−5.600429	13.32905
Carbon sink	−809,903.3	352,121.1	−2.30	0.07	−1,715,059	95,252.78
Technical investment	−270.6548	93.65129	−2.89	0.034	−511.3931	−29.91651
Carbon emission	−46,091.57	26,569.17	−1.73	0.143	−114,389.8	22,206.67
_cons	−4.83 × 10^7^	1.85 × 10^7^	−2.61	0.047	−9.57 × 10^7^	−813,838.1

**Table 12 ijerph-20-01423-t012:** Criteria for the classification of carbon emission economic benefits and low-carbon development potential.

Level	Criteria for the Classification
High economic efficiency, high low-carbon potential	U > u, F > f
High economic efficiency, low carbon potential	U > u, F < f
Low economic benefits, high low-carbon potential	U < u, F > f
Low-carbon benefits, low- carbon potential	U < u, F < f

**Table 13 ijerph-20-01423-t013:** Analysis of the economic benefits of marine fishery carbon emissions and the development potential of low-carbon fisheries in nine coastal provinces.

Area	Economic Benefits		Low Carbon Potential	
	Year 2009	Year 2019	Year 2009	Year 2019
Hebei	Low	-	Low	↑
Liaoning	Low	-	High	-
Jiangsu	High	-	High	↓
Zhejiang	Low	↑	Low	-
Fujian	High	↓	High	-
Shandong	High	-	High	-
Guangdong	Low	-	Low	-
Guangxi	Low	-	High	-
Hainan	High	-	Low	-

↑ means that 2019 has increased compared with 2009. - means that 2019 will remain the same as 2009. ↓ means that 2019 is lower than 2009.

**Table 14 ijerph-20-01423-t014:** Decoupling and coupling relationship analysis table of nine coastal provinces from 2010 to 2019.

Year	T_it_	State	Type of Development	T_io_	State	Type of Development
2010	0.35	Weak decoupling	Conventional	0.40	Weak coupling	Conventional
2011	0.09	Weak decoupling	Conventional	0.08	Weak coupling	Conventional
2012	0.04	Weak decoupling	Conventional	0.29	Weak coupling	Conventional
2013	0.19	Weak decoupling	Conventional	0.39	Weak coupling	Conventional
2014	0.82	Growth link	Conventional	0.71	Weak coupling	Conventional
2015	0.94	Growth link	Conventional	0.95	Growth coupling	Conventional
2016	0.36	Weak negative decoupling	Pollution type	0.64	Weak coupling	Conventional
2017	−7.38	Strong decoupling	Environmentally friendly	0.50	Weak coupling	Conventional
2018	−0.07	Strong decoupling	Environmentally friendly	0.12	Weak coupling	Conventional
2019	0.26	Weak negative decoupling	Pollution type	1.89	Strong coupling	Environmentally friendly

**Table 15 ijerph-20-01423-t015:** Analysis of the decoupling relationship of the nine coastal provinces.

	Year 2010			Year 2019		
Area	T_it_	State	Type of Development	T_it_	State	Type of Development
Hebei	−0.02	Strong decoupling	Environmentally friendly	0.98	Growth link	Conventional
Liaoning	0.12	Weak decoupling	Conventional	3.23	Recession decoupling	Conventional
Jiangsu	−0.20	Strong negative decoupling	Pollution type	0.15	Weak negative decoupling	Pollution type
Zhejiang	−0.14	Strong decoupling	Environmentally friendly	0.17	Weak decoupling	Conventional
Fujian	0.09	Weak decoupling	Conventional	−0.57	Strong negative decoupling	Pollution type
Shandong	0.31	Weak decoupling	Conventional	0.20	Weak negative decoupling	Pollution type
Guangdong	−0.15	Strong negative decoupling	Pollution type	−5.43	Strong decoupling	Environmentally friendly
Guangxi	2.68	Expansion Negative Decoupling	Pollution type	−0.07	Strong decoupling	Environmentally friendly
Hainan	1.52	Expansion Negative Decoupling	Pollution type	−1.02	Strong decoupling	Environmentally friendly

**Table 16 ijerph-20-01423-t016:** Analysis of coupling relationship in nine coastal provinces.

	Year 2010			Year 2019		
Area	T_io_	State	Type of Development	T_io_	State	Type of Development
Hebei	0.37	Weak coupling	Conventional	−3.71	Strong back lotus root	Pollution type
Liaoning	1.65	Strong coupling	Environmentally friendly	0.64	Weak coupling	Conventional
Jiangsu	1.42	Strong negative coupling	Pollution type	−0.03	Strong negative decoupling	Pollution type
Zhejiang	0.30	Weak coupling	Conventional	0.78	Weak coupling	Conventional
Fujian	0.15	Weak coupling	Conventional	1.26	Strong coupling	Environmentally friendly
Shandong	0.13	Weak coupling	Conventional	2.70	Strong negative decoupling	Pollution type
Guangdong	0.21	Weak coupling	Conventional	0.10	Weak coupling	Conventional
Guangxi	0.63	Weak coupling	Conventional	0.31	Weak coupling	Conventional
Hainan	0.17	Weak coupling	Conventional	17.82	Strong negative decoupling	Pollution type

**Table 17 ijerph-20-01423-t017:** Classification standard of fishery economic development models.

Fishery Carbon Emission Economic Development Model	Types of Decoupling States of Fishery Carbon Emission Economic Development	Types of Decoupling States of Fishery Carbon Emission Economic Development
Ideal	Environmentally friendly	Environmentally friendly
	Environmentally friendly	Conventional
General	Conventional	Environmentally friendly
	Conventional	Conventional
Inferior	Environmentally friendly	Pollution type
	Pollution type	Environmentally friendly
	Pollution type	Conventional
	Conventional	Pollution type
	Pollution type	Pollution type

**Table 18 ijerph-20-01423-t018:** Development mode of fishery economy in nine coastal provinces in 2010 and 2019.

Area	Development Model		
	Year 2010	Year 2019	
Hebei	Ideal	Inferior	
Liaoning	General	General	
Jiangsu	Inferior	Inferior	
Zhejiang	Ideal	General	
Fujian	General	Inferior	
Shandong	General	Inferior	
Guangdong	Inferior	Ideal	
Guangxi	Inferior	Ideal	
Hainan	Inferior	Inferior	

## Data Availability

Not applicable.

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
