# Peer review of "Research on the Duality of China’s Marine Fishery Carbon Emissions and Its Coordination with Economic Development"

_ijerph, 2023, doi:10.3390/ijerph20021423_

Round 1
Reviewer 1 Report
This study evaluated the duality of marine fishery carbon emission and its coordination with economic development. This topic is interesting and important to the related research field. But there are improvement to be made before publication.
1. The manuscript should follow the scientific article structure. That is abstract, introduction, method, results and discussion, and conclusion. Section 2 literature should be included in introduction section. Section 4 &5 are results and discussion section.
2. Conclusion should be concise and short. The recommendation can be put in the separated section as Perspectives.
3. There are many careless mistakes for spelling or grammars, i.e. line 18, what is “conventional ones”; line 34-36, there are grammar mistake; line 38-45, are they one sentence or two? They are too long; line 103, what is the “deepening of the research perspective”. The language should be carefully checked and polished.
4. Some citation is not accurate, i.e. in line 52, it can be revised as Tang et al., believed that ….; line 55, Yue et al., concluded that; line 60, Ziegler et al., pointed out that, there are also such mistakes in other places. Please carefully check and revise them.
Author Response
Response to Reviewer Comments
Dear editors and reviewers,
Thank you very much for your comments and suggestions.
Those comments are all valuable and very helpful for revising and improving our paper, as well as the important guiding significance to our researches. We have studied comments carefully and have made correction which we hope meet with approval. We have revised the paper in detail where readers may be confused. The changed parts of the article are marked in red. The main corrections in the paper and the responds to the reviewer’s comments are as following:
Reviewer1:
Point 1: The manuscript should follow the scientific article structure. That is abstract, introduction, method, results and discussion, and conclusion. Section 2 literature should be included in introduction section. Section 4 &5 are results and discussion section.
Response 1: Thank you for your valuable comments.We have restated it.
According to your comments, the structure of this paper is divided into six parts: introduction, data sources and methods, dual analysis of marine fisheries, coordination analysis between carbon emissions of marine fisheries and economic development of marine fisheries, conclusions and recommendations. The first part is introduction and literature review, the second part is data sources and methods, the third and fourth parts are data analysis for research problems, the fifth part is conclusion, and the sixth part is suggestions.
Point 2: Conclusion should be concise and short. The recommendation can be put in the separated section as Perspectives.
Response 2: Thank you for your valuable comments.We have restated it.See the article for details.
In this paper, the carbon emissions and carbon sinks of marine fisheries in nine coastal provinces from 2009 to 2019 are analyzed, and the decoupling relationship between carbon emissions and carbon sinks of marine fisheries and economic development is obtained by combining the changes of economic development in nine coastal provinces. Finally, the following conclusions are drawn:
(1) From the changes of marine fishery carbon emissions, it is known that the carbon emissions of nine coastal provinces are obviously different. The carbon emissions of marine fishery in Zhejiang ranks first, and the carbon emissions of marine fishery in other areas except Zhejiang are less than 3 million tons. From the change of marine fishery carbon sink, it can be seen that the highest marine fishery carbon sink is in Shandong, with an average of 300,000 tons, followed by Fujian > Guangdong > Liaoning > Zhejiang > Guangxi > Jiangsu > Hebei > Hainan. The net carbon emission of fishery shows that the net carbon emission of Zhejiang is higher than that of other regions, and that of Hebei is the lowest. Zhejiang has a long history of fishery development, and Zhoushan Fishing Ground is located here. The base of fishing boat operation is ahead of other provinces, and the scale of marine fishing is large. There is much room for improvement in terms of reducing emissions and increasing foreign exchange, or improving fishery technology.
(2) From the data of economic benefits of carbon emissions from marine fisheries, it is known that the economic benefits of carbon emissions from Jiangsu, Fujian, Shandong and Hainan are at a high level, which indicates that the economic benefits per unit of carbon emissions are high, and the carbon emissions from marine fisheries are high at the same time of economic development. From the change of low-carbon development potential, it can be seen that the low-carbon development potential of Liaoning, Jiangsu, Fujian, Shandong and Guangxi is at a high level.
(3) According to the calculation of marine fishery carbon emissions and carbon sinks combined with the economic GDP data of nine coastal provinces, the decoupling model of carbon emissions, carbon sinks and economic development is constructed, the decoupling index is calculated and the decoupling relationship is analyzed. The decoupling values of Hebei, Zhejiang and Guangdong are low, which indicates that these regions have taken the environment into consideration while developing their economies, and are realizing low-carbon economic development. Overall, the decoupling relationship between marine fishery carbon emissions and economic development in nine coastal provinces is obvious, and the decoupling state is relatively good.
6.Recommendations
Nine coastal provinces are in the rising stage of social and economic development, and carbon dioxide emissions will inevitably occur in the process of development. This study analyzes the decoupling relationship between carbon emissions, carbon sinks and economic development in marine fisheries, and on the basis of the research results, combined with the actual situation of industrial economy in various regions, puts forward measures and suggestions for reducing emissions and increasing sinks and developing green low-carbon economy from the following aspects.
(1) Increase shellfish culture, systematically optimize the variety structure of shellfish, and promote the promotion of ecological and economic benefits of marine fisheries. At present, the ecological and economic benefits of China's marine fisheries need to be improved due to the single breed and unreasonable structure of aquaculture. For a long time, the absolute quantity and growth rate of shellfish culture are the largest. In order to improve the comprehensive benefits of marine fishery eco-economy in China. Under the condition of basically meeting the demand, we should pay attention to the adjustment, optimization and promotion of mariculture variety structure to realize the sustainable growth of marine fishery.
(2) Improve the carbon emission policy and promote the development of carbon emission trading. Using policy and legal means to formulate binding indicators and management regulations of marine fishery carbon emissions can ensure the smooth implementation of low-carbon utilization management objectives of marine fishery, and is conducive to the smooth development of low-carbon work of marine fishery. Nine provinces should improve the carbon trading market system, strengthen the construction of carbon trading system, and define the main scope of carbon emissions.
(3) The formulation of marine fishery economic policy should consider the local industrial structure of marine fishery. The industrial structure of marine fishery in each region is different, and the low-carbon development potential is also different. Promote the marketization process of marine fishery carbon sink, speed up the development of marine fishery economy, and make China marine fishery carbon sink a new economic growth point. Among the nine coastal provinces, choose the provinces with strong carbon sink capacity, carry out the marine carbon sink trading pilot, and build a modern economic system of marine fisheries.
Point 3: There are many careless mistakes for spelling or grammars, i.e. line 18, what is “conventional ones”; line 34-36, there are grammar mistake; line 38-45, are they one sentence or two? They are too long; line 103, what is the “deepening of the research perspective”. The language should be carefully checked and polished.
Response3: I'm sorry that the expression in the article caused you confusion. Thank you for your valuable comments.We have restated it.
- Line 18, We have made the following modifications:China's marine fishery economic development is still dominated by conventional types.
Table15. Classification standard of fishery economic development model
|
Fishery carbon emission economic development model |
Types of decoupling states of fishery carbon emission economic development |
Types of decoupling states of fishery carbon emission economic development |
|
Ideal |
Environmentally friendly |
Environmentally friendly |
|
Environmentally friendly |
Conventional |
|
|
General |
Conventional |
Environmentally friendly |
|
Conventional |
Conventional |
|
|
Inferior |
Environmentally friendly |
Pollution type |
|
Pollution type |
Environmentally friendly |
|
|
Pollution type |
Conventional |
|
|
Conventional |
Pollution type |
|
|
Pollution type |
Pollution type |
- line 34-36, there are grammar mistake:We revised the sentence,as follows:
The National "Fourteenth Five Year Plan" clearly proposed the concept of optimizing the offshore green aquaculture layout, building marine ranching, and developing sustainable pelagic fisheries.Under the background of vigorously developing low-carbon marine economy, marine fishery is an important part of marine economy, and its sustained and healthy development is crucial.
- line 38-45, are they one sentence or two? Two sentences, We have revised them as follows:
Marine fisheries have dual characteristics of "carbon source" and "carbon sink" in production activities. According to the definition of carbon source and carbon sink in the United Nations Framework Convention on Climate Change, the fuel consumption of fishing vessels engaged in marine fishing is an important source of carbon emissions in marine fisheries, namely "carbon source"[1], while "carbon sink" refers to the process and mechanism of promoting aquatic organisms to absorb carbon in water bodies through fishery production activities, and removing the carbon that has been converted into biological products from water bodies through fishing. Fishery production activities with carbon sink function usually do not need bait, such as shellfish and algae cultivation in marine fishery[2].
- Line 103 is amended as follows:With China's increasing support for the economic development of marine fisheries and the deepening of the integration of marine fisheries with other industries, carbon emission from marine fisheries has become an important content of emission reduction.
Point 4: Some citation is not accurate, i.e. in line 52, it can be revised as Tang et al., believed that ….; line 55, Yue et al., concluded that; line 60, Ziegler et al., pointed out that, there are also such mistakes in other places. Please carefully check and revise them.
Response 4: Thank you for your valuable comments.We have restated it.
According to your comments of the reviewer, We have made corresponding modifications in the article. See the article modification section for details.
Reviewer 2 Report
The manuscript has been analyzed with very interesting questions. The topic is original and addresses a specific gap in the field of marine fishery carbon emissions and economic development. The manuscript has to be improved in the abstract, introduction, and conclusion. In the abstract and introduction, it has to add the subject, research objectives, hypotheses, and expected results. The conclusions are consistent with the evidence and arguments presented. But, in conclusion, it has to add the main contribution and limitations of the study. The references are appropriate. The tables and figures are correct and are consistent with the results presented.
Author Response
Point 1: In the abstract and introduction, it has to add the subject, research objectives, hypotheses, and expected results. The conclusions are consistent with the evidence and arguments presented. But, in conclusion, it has to add the main contribution and limitations of the study. The references are appropriate. The tables and figures are correct and are consistent with the results presented.
Response 1: Thank you for your valuable comments.We have restated it.Part of the introduction is amended as follows:
The above research results provide theoretical basis and reference significance for this paper. However, due to the wide range of marine fisheries, it is difficult to unify accounting methods and the accuracy of data is not high. With China's increasing support for the economic development of marine fisheries and the deepening of the integration of marine fisheries with other industries, carbon emission from marine fisheries has become an important content of emission reduction. Moreover, there is a typical difference between marine fishery and other industries. It has both a large number of emission sources and a large number of absorption sources. It is a dual industry. Therefore, how to reduce the emission sources and increase the carbon sink of marine fisheries in emission reduction has become a problem worth studying. At the same time, due to China's vast territory, the development of fishery economy in different regions is quite different. Studying the duality of fishery economy and the coordination of fishery economy development in different regions will provide a basis for summarizing the development mode of fishery economy and promoting the development direction of low-carbon fishery with regional differentiation.
Reviewer 3 Report
The ocean is the largest pool of carbon on Earth, and Marine fisheries are an important part of it. The paper calculates the carbon emissions and carbon sinks of marine fishery in nine coastal provinces of China from 2009 to 2019. Through the OECD decoupling index and Tapio decoupling elasticity coefficient, it measures the "decoupling and coupling" relationship between carbon emissions, carbon sinks and economic growth in nine coastal provinces in China, and objectively evaluates them in combination with the economic benefits of carbon emissions and low carbon development potential, Further, the development model of marine fishery economy is divided in detail and Some suggestions are put forward for the development of low carbon fishery, which has certain guiding significance. The quality and significance of the work presented in the manuscript is good, some novel conclusions are concluded. My suggestions are as follows:
1. Line 51, “The The research on carbon emissions” ?
2. Line 141-142 “Therefore, this paper uses this method to calculate the shell and algae carbon based on the relationship between the carbon sink coefficient, production and carbon sink. sink capacity.” ?
3. Whether the data in Table 2 are from the references, please indicate if so.
4. The sequence of figures and words in the text is out of order. Some are words before the corresponding figures, some are words after the corresponding figures.
5. The author has studied the Marine fisheries carbon sources, carbon sinks and their relationship with economic development in 9 provinces, and has come up with some results and put forward some suggestions. However, I am curious about the reasons behind these results and why there are these differences among different provinces. Is it because of the way ocean fisheries are caught? Or a difference in farming techniques? Perhaps a proper analysis of the reasons behind the results will make the recommendations more targeted.
Author Response
Reviewer 3:
Point 1:Line 51, “The The research on carbon emissions”
Response 1: Thank you for your valuable comments.We have restated it.
We have changed to“The research on carbon emissions”
Point 2: Line 141-142 “Therefore, this paper uses this method to calculate the shell and algae carbon based on the relationship between the carbon sink coefficient, production and carbon sink. sink capacity.”
Response 2: Thank you for your valuable comments.We have restated it.
“Therefore, this paper uses this method to calculate the shell and algae carbon based on the relationship between the carbon sink coefficient, production and carbon sink. sink capacity.”Part 2.3 of the carbon sink measurement method has been described. After checking the article, we found that this sentence is not appropriate here, so we deleted it.
Point 3: Whether the data in Table 2 are from the references, please indicate if so.
Response 3: Thank you for your valuable comments.We have restated it.
The data in Table 2 are from references, which We have indicated. The calculation methods and accounting coefficients of carbon sinks of shellfish and algae were obtained by referring to Tang et al [3], Zhang et al [6], Shao et al [8], Yue et al[16], Ji et al [18](see Equation 2-6, Table2).
Point 4: The sequence of figures and words in the text is out of order. Some are words before the corresponding figures, some are words after the corresponding figures.
Response 4: Thank you for your valuable comments.We have restated it.See the article for details.
Point 5: The author has studied the Marine fisheries carbon sources, carbon sinks and their relationship with economic development in 9 provinces, and has come up with some results and put forward some suggestions. However, I am curious about the reasons behind these results and why there are these differences among different provinces. Is it because of the way ocean fisheries are caught? Or a difference in farming techniques? Perhaps a proper analysis of the reasons behind the results will make the recommendations more targeted.
Response 5: Thank you for your valuable comments.We have restated it.
Shandong Province is a major marine fishery province in China and one of the important fishery production bases in China. The output of kelp culture accounts for more than 50% of the total aquaculture output in China. In February, 2011, the State Council officially approved the Plan of Zhejiang Marine Economic Development Demonstration Zone, which marked that the construction of Zhejiang Marine Economic Development Demonstration Zone was promoted to a national strategy. As a result, the development of marine fishery economy in Zhejiang Province entered the fast lane, made great progress, and played a leading role in the whole country. Fujian Province benefited from the national policy earlier. In 2010, it implemented preferential measures of fishery policy insurance, such as increasing the insurance amount of fishing boats, increasing financial subsidies, and expanding the scope of pilot projects, which promoted the expansion of mariculture area in Fujian Province, enhanced its carbon sequestration capacity, and its economic value was at the forefront. The planning and implementation of Guangdong Marine Economy Comprehensive Experimental Zone in 2012 laid a policy foundation for the development of marine fishery economy.
Reviewer 4 Report
Taking nine coastal provinces in China as an example, this paper calculates the carbon emissions and carbon sinks of marine fisheries from 2009 to 2019. Based on the OECD decoupling index and Tapio decoupling elasticity coefficient, it further measures the "decoupling and coupling" relationship between carbon emissions, carbon sinks and economic growth. In general, the article is not innovative enough.
1. The introduction does not give a clear explanation of the problem, so it is necessary to further strengthen the elaboration of the research significance of the paper and clarify the innovation.
2. The article lacks the empirical research on the factors that affect the sustainable development of China's marine fishery. I suggest that the authors refer to the current research on the factors affecting the sustainable development of the mining and metal industries and carry out in-depth empirical analysis, so as to draw more practical conclusions.
Author Response
Point 1: The introduction does not give a clear explanation of the problem, so it is necessary to further strengthen the elaboration of the research significance of the paper and clarify the innovation.
Response 1: Thank you for your valuable comments.We have restated it.
The above research results provide theoretical basis and reference significance for this paper. However, due to the wide range of marine fisheries, it is difficult to unify accounting methods and the accuracy of data is not high. With China's increasing support for the economic development of marine fisheries and the deepening of the integration of marine fisheries with other industries, carbon emission from marine fisheries has become an important content of emission reduction. Moreover, there is a typical difference between marine fishery and other industries. It has both a large number of emission sources and a large number of absorption sources. It is a dual industry. Therefore, how to reduce the emission sources and increase the carbon sink of marine fisheries in emission reduction has become a problem worth studying. At the same time, due to China's vast territory, the development of fishery economy in different regions is quite different. Studying the duality of fishery economy and the coordination of fishery economy development in different regions will provide a basis for summarizing the development mode of fishery economy and promoting the development direction of low-carbon fishery with regional differentiation.
Point 2: The article lacks the empirical research on the factors that affect the sustainable development of China's marine fishery. I suggest that the authors refer to the current research on the factors affecting the sustainable development of the mining and metal industries and carry out in-depth empirical analysis, so as to draw more practical conclusions.
Response 2:Thank you for your valuable comments.We have restated it.
3.4 OLS Regression Study of Carbon Sink on Sustainable Development of Marine Fishery Economy
Table8 Decriptive statistics
|
|
(1) |
(2) |
(3) |
(4) |
(5) |
|
Variables |
N |
mean |
sd |
min |
max |
|
Economic value |
99 |
802.7 |
467.2 |
113.8 |
1,786 |
|
Carbon emission |
99 |
203.3 |
129.5 |
41 |
502 |
|
Carbon sink |
99 |
12.95 |
11.00 |
0 |
39 |
Table9 Decriptive statistics
|
Variables |
Economic value |
|
Carbon sink |
252,832.655*** |
|
|
(11.86) |
|
Carbon emission |
3,344.542 |
|
|
(1.39) |
|
Constant |
4069054.191*** |
|
|
(4.26) |
|
Observations |
99 |
|
R-squared |
0.394 |
|
F test |
0 |
|
r2_a |
0.382 |
|
F |
77.30 |
(Robust t-statistics in parentheses *** p<0.01, ** p<0.05, * p<0.1)
According to the data in Table 8, the average carbon emission of marine fisheries in nine coastal provinces is 2.033 million tons, with a minimum of 410,000 tons and a maximum of 5.02 million tons. The average value of marine carbon sink is 129,500 tons, and the maximum value is 390,000 tons. According to the data in Table 9, the economic development of marine fisheries is affected by carbon emissions and sinks of marine fisheries.
Round 2
Reviewer 1 Report
There are many spelling mistakes. The citation format is not correct, i.e. Chen et al., 2021, also it shows many (Error! Reference source not found.) Please careful checking the whole manuscript before submission.
Author Response
Point 1: There are many spelling mistakes. The citation format is not correct, i.e. Chen et al., 2021, also it shows many (Error! Reference source not found.) Please careful checking the whole manuscript before submission.
Response 1: Thank you for your valuable comments.We have revised it.See the article for details.
Reviewer 4 Report
Although the study has been improved, there are still some problems:
1. 3.4, the coefficient of OLS regression result is too large, and the authors need to deal with it.
2. Table8, since it is to explore the influencing factors of Marine fishery economic development, it is obviously not enough to only consider carbon sink and carbon emission factors, but also need to consider regional environmental regulation, marketisation, technical innovation and other factors. I suggest the authors refer to the existing literature on exploring the impact of relevant factors on sustainable development and strengthen this part.
